# Biosynthesis of Rubellins in *Ramularia collo-cygni*—Genetic Basis and Pathway Proposition

**DOI:** 10.3390/ijms23073475

**Published:** 2022-03-23

**Authors:** Francois Dussart, Dorota Jakubczyk

**Affiliations:** 1Department of Agriculture, Horticulture and Engineering Science, Scotland’s Rural College (SRUC), Edinburgh EH9 3JG, UK; 2Institute of Bioorganic Chemistry, Polish Academy of Sciences, 61-704 Poznań, Poland

**Keywords:** anthraquinone biosynthesis, Baeyer–Villiger monooxygenase, polyketide synthase, *Ramularia collo-cygni*, rubellin, chrysophanol, secondary metabolite

## Abstract

The important disease Ramularia leaf spot of barley is caused by the fungus *Ramularia collo-cygni*. The disease causes yield and quality losses as a result of a decrease in photosynthesis efficiency due to the appearance of necrotic spots on the leaf surface. The development of these typical Ramularia leaf spot symptoms is thought to be linked with the release of phytotoxic secondary metabolites called rubellins in the host. However, to date, neither the biosynthetic pathways leading to the production of these metabolites nor their exact role in disease development are known. Using a combined in silico genetic and biochemistry approach, we interrogated the genome of *R. collo-cygni* to identify a putative rubellin biosynthetic gene cluster. Here we report the identification of a gene cluster containing homologues of genes involved in the biosynthesis of related anthraquinone metabolites in closely related fungi. A putative pathway to rubellin biosynthesis involving the genes located on the candidate cluster is also proposed.

## 1. Introduction

Ramularia leaf spot (RLS) is an important disease of barley caused by the ascomycete fungus *Ramularia collo-cygni*. The disease was originally described in Europe in 1893 [1] but has become a concern in the past 20 years as it became more prevalent and widespread. The fungus is now present in many temperate regions of the world, including Australia, New Zealand [2], Argentina [3], South Africa [4], Russia, and North America [5,6]. The disease is marked by the appearance of typical rectangular necrotic spots on leaves. In severe epidemy, symptoms can then develop on other aerial parts of the plant such as stem, awns, and grains [5]. The development of symptoms causes yield losses that can exceed 70% in extreme cases but are typically in the range of 10 to 20% [6]. Furthermore, the disease can cause quality losses as grain size, a trait used in the malting industry to assess grain quality, can be reduced by up to 4% [6]. Controlling RLS outbreaks has become increasingly difficult because of the withdrawal in many countries of multisite chemicals for environmental reasons [7]. In addition, the fungus has now developed resistance to all major classes of single-site fungicides such as quinone outside inhibitors (QoI) [8], succinate dehydrogenase inhibitors (SDHI), and sterol demethylation inhibitors (DMI) [9,10,11,12]. To date, no source of genetic resistance has been identified in barley; however, the genetic background of the host is known to play a role in disease development as varieties carrying the *mildew locus O* (*mlo*) mutation, which confers resistance to powdery mildew, exhibit increased susceptibility to RLS [13]. Further supporting the role of the host genetic background in RLS development, a loss of function mutation of the *Enhanced Magnaporthe 1* (*emr1*) gene in a *mlo* background mitigates the effect of the *mlo* mutation on RLS severity leading to decreased RLS symptoms [14].

*R. collo-cygni* is often described as an endophyte that becomes necrotrophic under specific conditions leading to the development of RLS symptoms. Although the reason for the fungus transitioning from endophytic to necrotrophic is currently unknown, it has been linked with external stresses such as flowering or senescence [15]. Environmental conditions are also known to play an important role in RLS development as plants subjected to waterlogging exhibit increased RLS symptoms compared with plants subjected to a normal watering regime [16]. Conversely, drought has recently been shown to reduce RLS severity in the field [17]. At the cell level, the development of symptoms is thought to be associated with the induction of plant cell death through the release and action in the host of the rubellin phytotoxins (Figure 1) produced by *R. collo-cygni*. The involvement of a phytotoxin in RLS development was first hypothesised by Sutton and Waller following the observation that plant cell death was occurring away from the fungal hyphae [18]. The identification of rubellins in infected barley leaves and the observation that rubellins are able to induce the production of the reactive oxygen species (ROS), such as superoxide (O_2_^•−^) in vitro [19], further supported a putative role for the toxins in RLS development as ROS were reported at the site of symptoms development [16]. Furthermore, the breakdown of the antioxidant system, which detoxifies ROS via specialised enzymes such as catalases and superoxide dismutase, is known to occur during senescence which has also been linked with symptoms development. However, to date, the exact mode of action of these secondary metabolites (SMs) in planta remains unclear, and their role in RLS development has not yet been confirmed.

Although rubellins were mostly studied in relation to the development of RLS, these toxins were first identified in *Mycosphaerella rubella,* the agent responsible for a necrotic leaf spot disease of *Angelica silvestris* [20]. Rubellins, which comprise several derivatives, are a family of anthraquinone-derived metabolites. Only two derivatives named rubellin A **1** and B **2** were originally reported [20]. Four more derivatives named rubellins C **3**, D **4**, E **5**, F **6,** and 14-dehydro rubellin D **9** have since been identified in *R. collo-cygni* [21]; however, rubellins B **2** and D **4** were the most abundant in infected barley leaves [22] suggesting that these two metabolites might be the end product of the rubellin biosynthetic pathway. Using labelled acetate, Miethbauer et al. showed that similar to most anthraquinones, rubellins are synthesised through the polyketide pathway [21]. Furthermore, rubellins are structurally related to other anthraquinone-derived metabolites such as cladofulvin and dothistromin that are synthesised by nonreducing polyketide synthases (NR-PKSs) in the dothideomycete fungi *Cladosporium fulvum* and *Dothistroma septosporum*, respectively [23,24]. Therefore, the biosynthetic gene cluster leading to rubellin production is expected to be based on an NR-PKS core gene and to contain homologues of genes involved in the biosynthesis of other anthraquinone metabolites such as emodin, monodictyphenone or chrysophanol. These metabolites are intermediate in the biosynthesis of known anthraquinone-derived SMs such as cladofulvin, sterigmatocystin, and agnestin [23,25]. Dussart et al. identified several gene clusters located around polyketide synthases (PKSs) in the genome of *R. collo-cygni* [26]. Here we investigated the cluster located around the *Pks1* gene in the context of rubellin biosynthesis.

The aim of this study was to identify a candidate rubellin biosynthetic gene cluster in *R. collo-cygni*. Understanding the biosynthesis of these SMs could contribute to a better understanding of their role in *R. collo-cygni* biology and disease symptoms development which could help identify breeding targets to improve host resistance to RLS. Furthermore, in the light of the renewed interest in the antibiotic and antitumor activity of the rubellins and the potential of these SMs as therapeutics to treat Alzheimer’s disease through the inhibition of tau aggregates formation [27,28,29], understanding the biosynthesis of these SMs could provide an efficient and inexpensive source of rubellin. Currently, rubellin biosynthesis relies on synthetic chemistry [30]; therefore, unravelling the biosynthetic pathway in *R. collo-cygni* could pave the way for pathway engineering and ectopic expression of the cluster.

## 2. Results

### 2.1. Phylogenetic Analysis of R. collo-cygni Nonreducing-Polyketide Synthases

A previous study based on phylogenetic analysis and domain organisation of all *R. collo-cygni* PKSs showed that the genome of *R. collo-cygni* encodes three NR-PKSs that have been named Pks1, Pks2, and Pks3 [26]. To assess the putative role of the NR-PKSs identified in *R. collo-cygni*, a new phylogenetic study of the proteins RccPks1, RccPks2, and RccPks3 encoded by *Pks1*, *Pks2*, and *Pks3*, respectively, was carried out (Figure 2). RccPks3 was recovered on the same clade as MpaC and PksAC, the NR-PKSs responsible for the biosynthesis of the backbone of mycophenolic acid and aschochitine in *Penicillium brevicompactum* and *Aschochyta fabae*, respectively [31,32]. Two NR-PKSs involved in the biosynthesis of terpenoids, also located in the same clade as Trt4 and AndM, are involved in the biosynthesis of terretonin and anditomin in *Aspergillus terreus* and *A. stellatus*, respectively [33,34]. RccPks2 was recovered in a clade that contains enzymes involved in the biosynthesis of perylenequinones such as Pks1 in *Elsinoë fawceti* and CpPks1 in *Cladosporium phlei* that are responsible for the production of elsinochrome and phleichrome pigments, respectively [35,36]. RccPks2 is also located on the same clade as two NR-PKSs of unknown function in the closely related fungi *Zymoseptoria tritici* and *Pseudocercospora fijiensis*. RccPks1 was recovered on the same clade as ClaG and EncA, the enzymes responsible for the production of the anthraquinone-derived SMs cladofulvin and endocrocin in *C. fulvum* and *Aspergillus fumigatus*, respectively [23,37]. The NR-PKSs AgnPks1, Acas and MdpG that are involved in the biosynthesis of the anthraquinone SMs agnestin, atrochrysone and monodictyphenone, in *Paecilomyces divaricatus*, *Lasiodiplodia theobromae* and *A. nidulans*, respectively, were also located on the same clade as RccPks1 [25,38,39].

### 2.2. Description of the Candidate Rubellin Biosynthetic Gene Cluster

A gene cluster around the *Pks1* gene was previously identified by Dussart et al., and it exhibits synteny with the monodictyphenone cluster in *A. nidulans* [26]. Based on the phylogenetic study carried out here, the *Pks1* gene will hereinafter be renamed *rccG* as it exhibits similarity with *ClaG* and *MdpG*. The AntiSMASH [40] online genome analysis tool was interrogated and combined with an in silico genome walking approach to identify a more comprehensive candidate gene cluster that could be involved in rubellin biosynthesis. A protein–protein basic local alignment search tool (BLASTp) [41] analysis was then carried out to confirm homology with genes in known SMs clusters based on reciprocal BLAST. These analyses reveal that several genes in the *rccG* cluster also exhibit homology with genes involved in cladofulvin biosynthesis in *C. fulvum* as well as with genes involved in the monodictyphenone biosynthetic cluster (Figure 3). Furthermore, two new miniclusters were identified on Contig113 and Contig 280. The minicluster on Contig113 contains two genes; one gene is homologous to the Baeyer–Villiger monooxygenases (BVMOs) *claL* and *mdpL* involved in the biosynthesis of cladofulvin and monodictyphenone, respectively, while the other gene is a putative major facilitator superfamily (MFS) transporter. The minicluster located on Contig 280 comprises three genes, including a putative nonribosomal peptide synthetase (NRPS), a putative MFS transporter, and a monooxygenase homologous to *mdpD* in the monodictyphenone biosynthetic cluster. These two miniclusters will be named the *rccL-* and *rccD* miniclusters in the rest of the manuscript. Seven genes located on the *rccG* cluster exhibit homology with genes in both the cladofulvin and monodictyphenone biosynthetic clusters. Similarly, the *rccL* gene located in the *rccL* minicluster also exhibits homology to genes present in both cladofulvin and monodictyphenone clusters. A further three genes located on the *rccG* cluster exhibit homology only with genes found in the cladofulvin biosynthetic cluster, whereas the *rccD* gene exhibit homology only with the *mdpD* gene found on the monodictyphenone cluster.

### 2.3. Expression Profiling of Selected Genes in the Putative Rubellin Biosynthetic Clusters

To confirm the involvement of the genes identified in the candidate clusters, their expression profile was investigated. Coregulation being a sine qua non condition to be part of a given cluster, the previously published RNA-seq data [43] were interrogated to quickly identify genes in the candidate cluster that exhibited a similar expression pattern to that of the core PKS gene *rccG*. Of the 15 genes present alongside *rccG*, only nine showed coregulation based on the RNA-seq data (Figure 4). Both genes located on the *rccL* minicluster were expressed at the same time as *rccG,* and only *rccD* in the *rccD* minicluster was expressed at the same time as *rccG*. The expression of the putatively coregulated genes was further investigated during disease development in artificially inoculated barley seedling via qRT–PCR. Gene-specific primers exhibiting sufficient amplification efficiency were successfully designed for the nine genes present alongside *rccG*, for *rccD* as well as for both genes present on the *rccL* minicluster. Additionally, a primer set was also designed for a gene (*rcc_01325*) previously identified as sitting out with the cluster to confirm the validity of the RNA-seq data. Transcript levels of the *rcc_01325* gene did not appear to follow that of the other genes in the cluster, therefore confirming the validity of the RNA-seq approach. All the genes investigated on the *rccG* cluster as well as both genes on the *rccL* minicluster and *rccD* exhibited a similar expression profile with transcript levels being the highest at 5 and 7 dpi, decreasing after 10 dpi, and remaining low until 21 dpi (Figure 4).

### 2.4. Comparison of Rubellin and Chrysophanol Toxicity

Chrysophanol is an intermediate in the biosynthesis of cladofulvin and is synthesised by *claG*, *claF,* and *claH*. Based on the homology and gene-expression study carried out above, chrysophanol **7** may also be an intermediate in the biosynthesis of rubellins, as *rccG, rccF,* and *rccH* are coregulated and exhibit homology to *claG*, *claF,* and *claH*, respectively. Furthermore, the presence of chrysophanol was also confirmed in the liquid culture of *R. collo-cygni* by mass spectrometry (Appendix A).

Therefore, the biological activity of both chrysophanol **7** and rubellin D **4** was investigated in planta. Rubellins being nonhost-specific toxins [19], leaves of *Arabidopsis thaliana* were infiltrated with the two compounds, and lesions size was measured after 24 h. Leaves infiltrated with a solution of rubellin D **4** (50 µM) exhibited significantly (*p* value < 0.001) larger lesions than leaves infiltrated with a solution of chrysophanol **7** at an equal concentration (Figure 5). Chrysophanol **7** at 50 µM exhibited a similar lesion size to that of the control (10 mM MgCl_2_). Infiltration with a concentration of up to 1 mM of chrysophanol **7** still produced significantly (*p*-value < 0.001) smaller lesions than infiltration with 50 µM of rubellin D **4,** suggesting that it exhibits much greater biological activity than chrysophanol **7**.

## 3. Discussion

### 3.1. Identification of the Putative Core Gene in Rubellin Biosynthesis

The phylogenetic analysis carried out on the three NR-PKSs previously identified in the genome of *R. collo-cygni* strongly supports the role of *RccG* as the core gene in the rubellin biosynthetic pathway, as previously suggested by Dussart et al. [26]. Both *Pks2* and *Pks3* are unlikely to be involved in rubellin biosynthesis. RccPks3, the protein encoded by *Pks3*, is related to NR-PKSs involved in the biosynthesis of meroterpenoids such as MpaC, Trt4, and AndM, responsible for the biosynthesis of mycophenolic acid, terretonin, and anditomin in *P. brevicompactum*, *A. terreus*, and *A. stellatus*, respectively [31,33,34]. Therefore, *Pks3* is likely to be involved in the biosynthesis of a meroterpenoid SM. The protein encoded by *Pks2* in *R. collo-cygni* appears to be related to proteins involved in the biosynthesis of secondary metabolites belonging to the perylenequinone family, which are structurally very different from anthraquinone. RccG appears to be closely related to other enzymes involved in the biosynthesis of anthraquinone-derived SMs, such as cladofulvin and monodictyphenone in *C. fulvum* and *A. nidulans*, respectively. Taken together, these results suggest that the gene encoding RccG in *R. collo-cygni* may be the best candidate for the early steps of rubellin biosynthesis. Although all the NR-PKSs present in the clade containing RccG are involved in diverse biosynthetic pathways, all these pathways appear to rely on the formation of common intermediates. Compounds such as atrochrysone carboxylic acid **10** or emodin **8** can be found during the biosynthesis of SMs structurally related to rubellins such as monodictyphenone, agnestins, and cladofulvin in *A. nidulans*, *P. divaricatus*, and *C. fulvum,* respectively [23,25,42]. Similarly, chrysophanol **7** and chrysophanol hydroquinone are intermediate in the biosynthesis of agnestins and cladofulvin, respectively. Based on structural similarity between rubellins and cladofulvin, cladofulvin being a dimer of nataloe emodin and rubellin being a dimer of chrysophanol **7**, the biosynthetic pathways of these two SMs are expected to share several intermediates. Interestingly, Miethbauer et al. [44] have previously shown the presence of chrysophanol **7** in the culture of *R. collo-cygni,* suggesting that this intermediate could be involved in the biosynthesis of rubellin.

### 3.2. Identification of the Candidate Rubellin Biosynthetic Candidate Cluster

Through the combined in silico genome walking and BLAST analyses carried out in the present study, a more comprehensive candidate gene cluster for rubellin biosynthesis in *R. collo-cygni* was identified. The putative rubellin biosynthetic cluster appears to involve the main cluster organised around the core PKS gene, *rccG*, and two miniclusters located elsewhere in the *R. collo-cygni* genome and organised around the *rccL* and *rccD* genes. This organisation is reminiscent of that of other biosynthetic gene clusters responsible for the biosynthesis of SMs structurally related to rubellins. Dothistromin, an anthraquinone-derived SM produced by *Dothistroma septosporum*, a pathogen of pine trees, is synthesised by a cluster comprising six small clusters spread on chromosome 12 and a single gene located on chromosome 11 [45]. Similarly, the organisation of the cladofulvin biosynthetic cluster resembles that of the candidate identified in this study, as it consists of the main cluster comprising most of the gene required for cladofulvin biosynthesis located around the core PKS gene *claG* supplemented by two miniclusters, each containing a single gene [23]. Furthermore, all the genes present in the cladofulvin biosynthetic cluster have a homologue in the candidate cluster in *R. collo-cygni*, be it in the main cluster or in the *rccL* minicluster (Figure 3). The *rccL* minicluster consists of two genes, including the *rccL* BVMO. BVMOs are enzymes that catalyse the formation of an ester from a ketone through the integration of an oxygen atom adjacent to the carbonyl group. Although BVMO activity is relatively common in fungi [46], only a limited number of BVMOs has been characterised in fungal genomes. *R. collo-cygni rccL* exhibits homology with several BVMOs that have been identified in gene clusters responsible for the biosynthesis of SMs structurally related to rubellins. Chiang et al. [42] reported that the BVMO *mdpL* was involved in the biosynthesis of monodictyphenone. Similarly, *rccL* also exhibit homology with *claL* and *agnL3*, the BVMOs involved in the biosynthesis of cladofulvin and agnestins in *C. fulvum* and *P. divaricatus*, respectively [23,25]. The *rccD* minicluster, which does not comprise homologues to the cladofulvin biosynthetic genes, likely consists of a single monooxygenase (*rccD*) as the organisation of the condensation, adenylation and peptidyl carrier protein domains of the putative NRPS identified in this minicluster suggests this protein is nonfunctional [26]. Furthermore, this NRPS and the MFS transporter located near *rccD* showed no expression based on the previously published RNA-seq study [43].

The full complement of genes involved in the candidate gene cluster was further defined via a gene expression approach. Ten genes located on the main *rccG* cluster, both genes on the *rccL* minicluster as well as the *rccD* gene on the *rccD* minicluster, appear to be coregulated (Figure 4). The transcript levels of these genes were highest during the asymptomatic and early symptoms formation stages of infection (5 and 7 dpi). Dussart et al. [26] reported similar findings for several PKSs, NRPSs, and hybrids PKS-NRPS in *R. collo-cygni*, suggesting that several SMs may be synthesised and involved in the early stages of disease development. These results contrast with those observed for the dothistromin biosynthetic cluster in *D. septosporum* as all the genes present in that cluster exhibit their highest expression level during the mid and late stages of disease development, i.e., during lesions formation and sporulation [47]. Similarly, the genes responsible for cladofulvin biosynthesis are downregulated during the biotrophic growth of *C. fulvum* in planta [48]. Dothistromin is a known virulence factor that mediates disease severity, whereas *C. fulvum* lines overproducing cladofulvin showed larger lesions compared with the wild type but reduced sporulation. In the closely related wheat pathogen *Zymoseptoria tritici*, the expression profile of SM gene clusters was also studied. Two clusters organised around PKSs exhibit a similar expression profile to that of the *R. collo-cygni rccG* cluster and were hypothesised to be involved in the conidial germination and stomatal penetration stage of disease development [49]. Six other clusters organised around PKSs and NRPSs showed the highest transcript levels during the mid and late stages of disease development and were associated with the initiation of tissue necrosis, host cell death, and sporulation. Considering that the rubellin candidate cluster is mostly expressed during the early stages of disease development, these results suggest that, unlike cladofulvin, rubellins may not be involved in sporulation but are more likely to play a role in mediating the host–pathogen interactions, be it through the induction of symptom formation or through the protection of an ecological niche against other pathogenic fungi.

### 3.3. Proposed Pathway for Rubellin Biosynthesis

The anthraquinone scaffold that forms the backbone of all rubellins has been previously suggested to be biosynthesised via a polyketide pathway, while the previously mentioned *rccG* NR-PKS in *R. collo-cygni* may be responsible for the biosynthesis of this anthraquinone scaffold (Figure 6). InterProScan analysis [50] showed the presence of a nonreducing, iterative type I PKS with MdpG- or ClaG-like domains: ACP—acyl carrier protein; PT—product template; AT—acyl transferase; KS—keto synthase and SAT—starter unit acyl carrier protein transacylase. The next step in the pathway that leads to atrochrysone carboxylic acid **10** could be catalysed by RccF, an MdpF-like β-lactamase, which exhibits hydrolase activity especially considering that no termination thioesterase domain is present in RccG. Subsequently, atrochrysone carboxylic acid **10** would undergo decarboxylation to chrysophanol **7** by the decarboxylase analogue RccH. A tetrahydroxynaphthalene reductase T4HN or the MdpC short-chain dehydrogenase (SDR) homologue, RccC, could then dearomatise one of the chrysophanol **7** units. Two chrysophanol **7** units could be conjugated by the unique cytochrome P450, RccM, which could be responsible for the dimerisation of these two anthraquinone units. The proposed candidate, RccM, exhibits 42% identity to ClaM, a fungal anthraquinone dimerisation P450 [23]; however, to the best of our knowledge, no double dimerising fungal oxidase is known to date. The putative rubellin biosynthetic pathway branches at this point in two directions leading to rubellins C **3** and D **4**. The anthraquinone unit of dimer **11** could be hydroxylated by RccD, a FAD-dependent monooxygenase-like protein. Both intermediates **11** and **12** would undergo oxidation to a seven-membered lactone ring by the rare BVMO, RccL. RccL shares 43% identity to other known BVMOs, such as MdpL as well as to AgnL3 in the agnestin biosynthetic pathway based on reciprocal BLAST (e-value < e^−99^) [42]. The last steps of both branches of this pathway may be spontaneous and lead to rubellin C **3**, D **4**, E **5,** and F **6**. It is important to note that the formation of previously observed 14-dehydro rubellin D **9** could be catalysed by the dehydratase-like protein, RccB.

### 3.4. Comparison of Rubellin and Chrysophanol Toxicity

The phytotoxicity of chrysophanol **7**, an intermediate in the biosynthesis of rubellins, was compared with that of rubellin D **4** (Figure 5). Chrysophanol **7** exhibited significantly lower phytotoxic activity than rubellin D **4** when infiltrated in leaves of *A. thaliana*. In the proposed pathway, rubellins, including rubellin D **4**, resulted primarily from the dimerisation of chrysophanol **7** (Figure 6). Dimerisation and the atomic rearrangement associated with it can profoundly alter the biological activity of a molecule. For instance, jesterone, an anti-oomycete epoxyquinone SM produced by the endophytic fungus *Pestalotiopsis jesteri* [51], inhibits the growth and development of cancer cells but a synthetic dimer of jesterone exhibits up to 100 times greater activity against the same cancer lines [52]. Similar findings were previously reported for SM structurally related to rubellins as the activity of nataloe-emodin was compared with that of cladofulvin. Cladofulvin, a dimer of nataloe-emodin, exhibited up to 320 times higher cytotoxic activity than nataloe-emodin when tested against different animal cell lines [23]. The cytotoxic and antibiotic activities of chrysophanol **7** were previously compared with that of rubellins and uredinorubellins, another chrysophanol dimer intermediate in the biosynthesis of rubellins. At a concentration of 10 µM, chrysophanol **7** and rubellin D **4** exhibited a similar level of cytotoxicity against the HT29 and J774A.1 lines, but interestingly, only rubellin D **4** showed cytotoxicity against the HIG-82 line [44]. Rubellin D **4** and uredinorubellins also exhibited antibiotic activity against several strains of *Streptococcus aureus,* including methicillin-resistant *S. aureus* (MRSA), whereas chrysophanol **7** did not appear to affect bacterial growth [44]. Taken together, these results suggest that the dimerisation step in the biosynthesis of rubellins alters the biological activity of the SM and may be critical to considerably increase phytotoxic activity. The role of rubellins in RLS disease development being currently unknown, the *rccM* gene, which is a candidate for the dimerisation step, would be a good target for gene silencing. Deletion of *rccM* would result in the accumulation of the easily identified compound chrysophanol **7**, which can be used as a control to confirm successful rubellin biosynthesis disruption. Furthermore, *rccM* deletion would also result in decreased phytotoxicity, therefore, allowing for a better understanding of the role played by rubellins in the host–pathogen interactions.

## 4. Materials and Methods

### 4.1. Phylogenetic Analysis of R. collo-cygni NR-PKSs

Protein sequences of NR-PKSs involved in the biosynthesis of characterised metabolites were used in this study. The full-length sequences of protein identified in previous studies by Kim et al. (2019) and Dussart et al. (2018) [26,32] were downloaded from the Genebank database (Appendix A). Sequences were aligned using the multiple sequence comparison by log expectation (MUSCLE) tool [53]. The phylogenetic tree was built using PhyML version 3.3 [54] with 100 bootstrap replicates and nodal support value cutoff set at 75%.

### 4.2. Identification of the Candidate Rubellin Biosynthetic Candidate Cluster

The *rccG* cluster had previously been described by Dussart et al. [26] using a combination of in silico genome walking combined with a BLAST approach. To refine this cluster and identify the *rccL* and *rccD* miniclusters, a similar combination of methods has been used as the antiSMASH database [40] was applied to interrogate the genome of *R. collo-cygni* to identify potential clusters of genes involved in secondary metabolism. Subsequently, an in silico genome walking as described by Dussart et al. [26] and BLASTp [41] analysis were used to confirm the presence of the two putative genes on the *rccL* minicluster as well as the three putative genes on the *rccD* minicluster. Finally, the homology between the genes in the candidate rubellin clusters was investigated through reciprocal BLAST against the monodictyphenone and cladofulvin biosynthetic gene clusters (Appendix A). The borders of both *rccL* and *rccD* miniclusters were delineated either by the end of the contig on which the genes are found or by three consecutive genes with a predicted function not involved in secondary metabolism.

### 4.3. Expression Profiling of Putative Rubellin Biosynthetic Genes

Transcript levels of candidate genes were assessed in artificially inoculated barley seedlings following the inoculation method developed by Makepeace et al. [55] and Peraldi et al. [56]. Plant material used to assess the transcript level of candidate rubellin biosynthetic genes was obtained from Dussart et al. [26]. Seedlings of the barley cultivar Century were grown for two weeks at 18 °C, 80% humidity, and 16 h of light (250 µmol m^−2^ s^−1^), then sprayed with a mycelial suspension obtained from a two-week-old *R. collo-cygni* isolate DK05 culture grown in potato dextrose broth (PDB). Infection was promoted by ensuring maximum humidity and incubating the plants for 48 h in the dark at 18 °C. After incubation, previously described growing conditions were restored with the exception of humidity levels which were maintained at the maximum level for a further 72 h before returning to 80%. Disease progression was visually assessed at 5, 7, 10, 12, 15, 18, and 21 days postinoculation (dpi). At each time point, two leaves were harvested and pooled per biological replicate and immediately snap-frozen in liquid nitrogen. *R. collo-cygni* DNA was quantified throughout the experiment and was previously reported in Dussart et al. [26]. RNA extraction was carried out using the RNeasy mini kit (Qiagen Ltd., Hilden, Germany) following the manufacturer’s instructions, and potential genomic DNA contaminants were removed using the TURBO DNase I kit (Thermo Fisher Scientific, Austin, TX, USA) following the manufacturer’s instructions. cDNA synthesis was carried out with the SuperScript III First-Strand Synthesis kit (Invitrogen, Carlsbad, CA, USA) following the manufacturer’s instructions prior to being used in qRT–PCR with a SybrGreen Jump Start Taq system (Sigma, Dorset, UK). The stability of four housekeeping genes was assessed, and the two most stable ones, glyceraldehyde-6-phosphate dehydrogenase (GAPDH) and α-tubulin, were used to normalise cDNA using GeNorm [57]. Using gene-specific primers (Appendix A) with amplification efficacies ranging from 80% to 110%, transcript quantification was carried out based on the E^−ΔΔCt^ method [58]. Data presented were mean normalised expression value ± standard error of three independent experiments.

### 4.4. Chrysophanol Detection

Liquid cultures of *R. collo-cygni* isolate I20 were grown in Czapek Dox medium for eight weeks in the dark at room temperature. The culture was filtered, and the aqueous phase was acidified to pH 4 and extracted three times with ethyl acetate, according to the previously published procedure [21]. The organic phase was dried over magnesium sulphate and evaporated to dryness. The dried extract was diluted with LCMS grade methanol, centrifuged, and filtered through a 0.22 µm filter.

Mass spectrometry detection was performed on a Shimadzu high-performance liquid chromatography (Shimadzu, Kyoto, Japan) quadrupole-type tandem mass spectrometer, LCMS-8040 instrument equipped with an electrospray (ESI) source operated in negative ion mode. The samples were eluted in a water (A)/acetonitrile (B) phases system in 80% of phase B, without any chromatographic column. The total flow was 0.300 mL/min.

The sample introduced from the liquid chromatography was sprayed and ionised under atmospheric pressure by the atmospheric pressure ionisation probe (ESI). The ionised sample was introduced through the sample introduction unit, a desolvation line (DL). The Heat Block temperature was 200 °C, and the DL temperature was kept at 200 °C. Nebulising gas flow was 3.00 L/min, drying gas flow was 15.00 L/min, and CID gas pressure was 230 kPa. The ions were separated according to their mass-to-charge ratio (*m/z*). The detected ion signals were processed by the LabSolutions data processing software. The observed mass of the chrysophanol **7** molecular ion was [M – H]^−^ = 253.10 in negative ion mode (Appendix A).

### 4.5. Comparison of the Phytotoxicity of Rubellin D and Chrysophanol

To compare the toxicity of rubellin D **4** with that of chrysophanol **7**, foliar infiltrations were carried out in the model plant *Arabidopsis thaliana* as described in Dussart et al. [59]. Stock solutions (10 mM) of rubellin D **4** (Enzo Life Sciences Farmingdale, Farmingdale, NY, USA) and chrysophanol **7** (Sigma Aldrich, Dorset, UK) were obtained by dissolving the pure compound in dimethyl sulfoxide (DMSO). Working solutions were prepared on the day of use by dissolving the stock solutions in 10 mM MgCl_2_ (Sigma Aldrich, Dorset, UK). Control solution consisted of a 10 mM MgCl_2_ solution supplemented with a volume of DMSO equivalent to that of the treatment. Working solutions were infiltrated using a 1 mL syringe on the abaxial side of the third or fourth leaf of *A. thaliana* plants ecotype Columbia-0 (Col-0) grown for two weeks at 18 °C, 16 h light photoperiod (250 µmol m^−2^ s^−1^) and 80% relative humidity. To ensure that the full leaf area was infiltrated, infiltrations were carried out at mid-length of the leaf on either side of the central vein. Following infiltrations, plants were incubated for 24 h in the previously described conditions before lesions were photographed and measured using the ImageJ software [60]. Data show average relative lesion size ± standard error based on three independent experiments, each consisting of 10 leaves per treatment. Data analyses were performed in R [61]. Analysis of variance (ANOVA) was used to show the effect of treatments on lesion size, and Tukeys HSD was used to show differences among treatments.

## Figures and Tables

**Figure 1 ijms-23-03475-f001:**
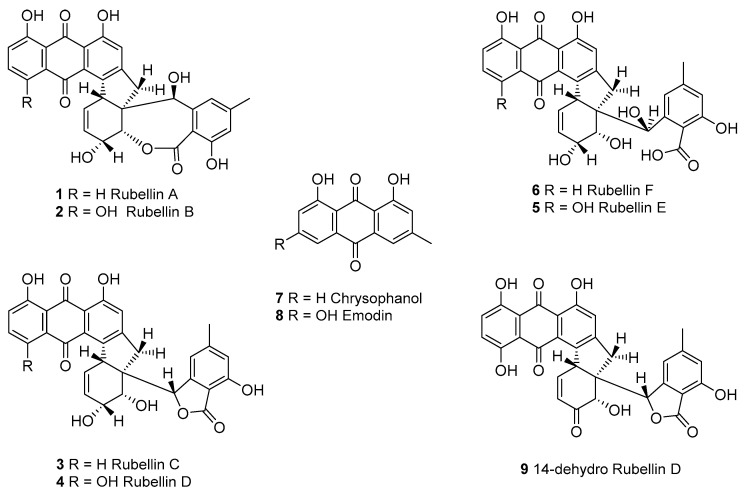
Structures of the rubellins A–F (**1**–**6**), chrysophanol **7**, the possible precursor of the rubellins and emodin **8**, its analogue, as well as the 14-dehydro rubellin D derivative **9**.

**Figure 2 ijms-23-03475-f002:**
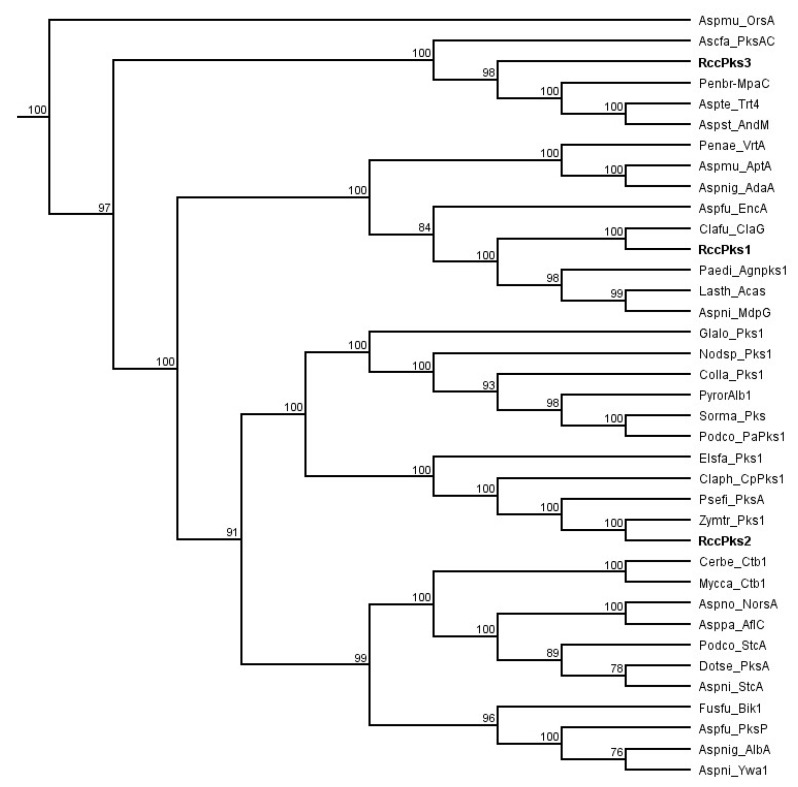
Phylogenetic analysis of *R. collo-cygni* nonreducing polyketide synthases (NR-PKSs). The cladogram was obtained using maximum likelihood on full-length protein sequences with 100 replications and nodal support value cutoff set at 75%. The list of sequences used in this study can be found in Appendix A.

**Figure 3 ijms-23-03475-f003:**
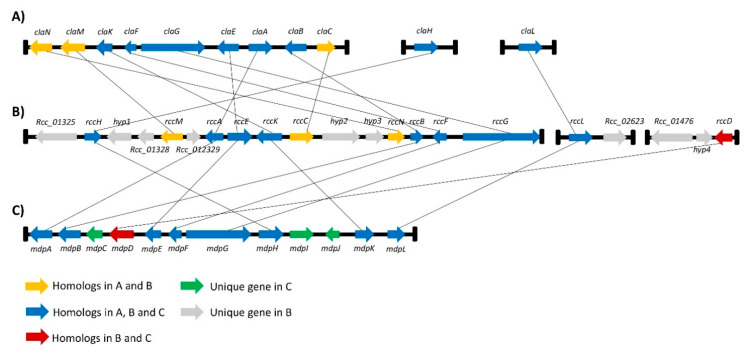
Homology between the rubellin candidate cluster and gene clusters involved in the biosynthesis of known secondary metabolites structurally related to rubellin; (**A**) The cladofulvin biosynthetic gene cluster in *C. fulvum* was described by Griffith et al.; (**B**) The rubellin candidate gene cluster in *R. collo-cygni* consists of gene clusters organised around the core gene *rccG* as well as two miniclusters named *rccL*- and *rccD* miniclusters organised around *rccL* and *rccD*, respectively; (**C**) The monodictyphenone biosynthetic gene cluster in *A. nidulans* was identified by Chiang et al. [42]. Homology was confirmed by reciprocal BLAST using the NCBI database.

**Figure 4 ijms-23-03475-f004:**
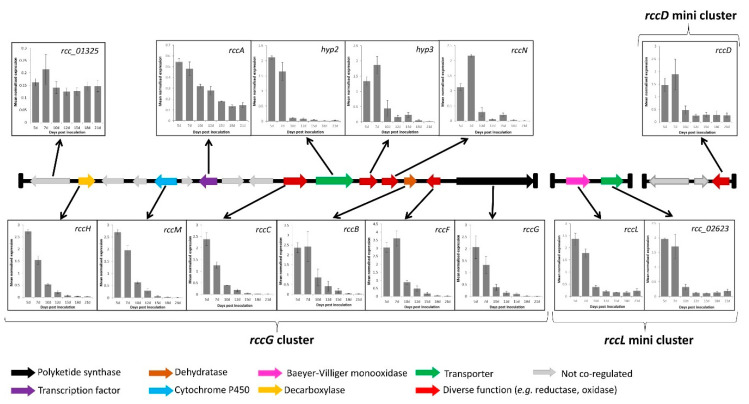
Expression of genes in the rubellin candidate biosynthetic cluster during disease development in artificially inoculated barley seedlings. Disease development was previously described by Dussart et al. [26]. Early symptoms appearance was recorded at 7 dpi, with typical lesions being clearly visible after 10 dpi. Genes represented in grey did not appear to be coregulated based on the RNA-seq data published by McGrann et al. [43]. The expression profile of gene *Rcc_01325* was used to validate the results of the RNA-seq data. Data show mean normalised expression value ± standard error of three independent experiments.

**Figure 5 ijms-23-03475-f005:**
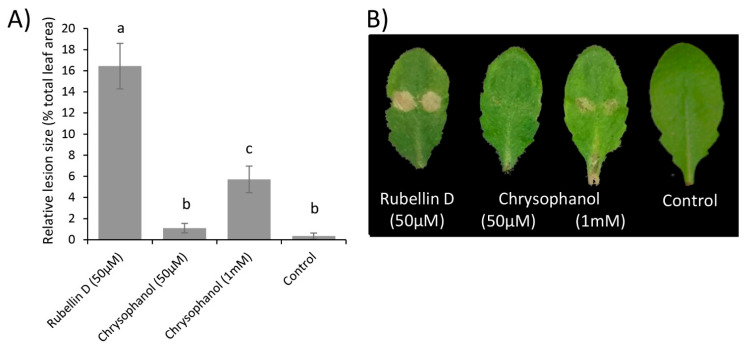
Lesions induced by rubellin D **4** and chrysophanol **7** following infiltration in leaves of *A. thaliana*. (**A**) Relative lesion size in leaves infiltrated with rubellin D **4**, chrysophanol **7,** or MgCl_2_ (control) measured after 24 h. Data show the average mean relative lesion size ± standard error of three independent experiments of at least 10 leaves each. Significant differences are shown by different letters. (**B**) Representative pictures of lesions observed after 24 h following infiltrations.

**Figure 6 ijms-23-03475-f006:**
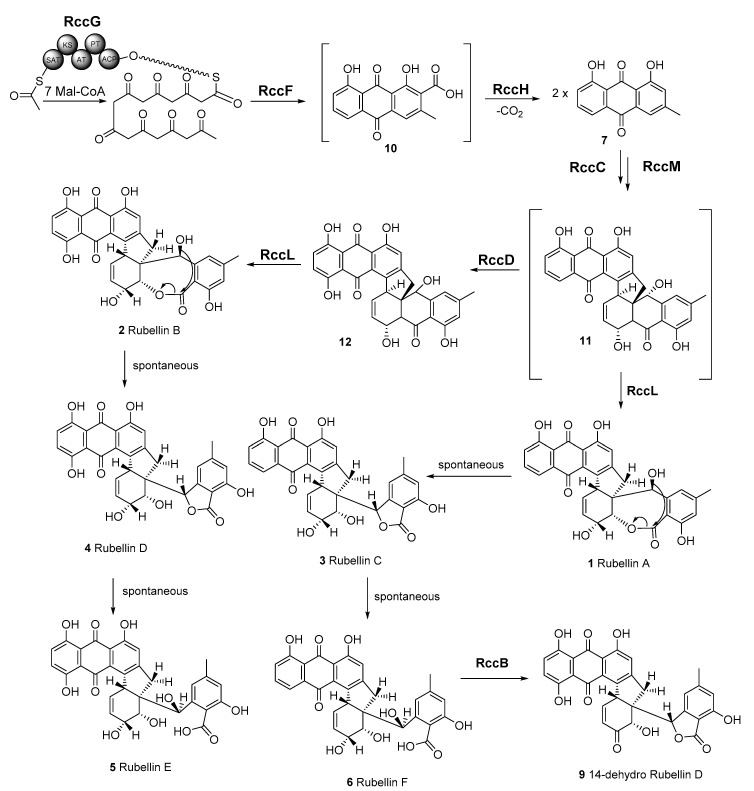
Proposed biosynthetic pathway of rubellins A–F (**1**–**6**) and 14-dehydro rubellin D **9**. RccG—nonreducing, iterative type I polyketide synthase (PKS). PKS domains: ACP—acyl carrier protein; PT—product template; AT—acyl transferase; KS—keto synthase; SAT—starter unit acyl carrier protein transacylase; RccF—β-lactamase; RccH—decarboxylase; RccC—short-chain reductase; RccM—cytochrome P450; RccD—FAD-dependent monooxygenase/hydroxylase; RccL—Baeyer–Villiger monooxidase; RccB—dehydratase.

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
