# Peer review of "Biosynthesis of Rubellins in Ramularia collo-cygni—Genetic Basis and Pathway Proposition"

_ijms, 2022, doi:10.3390/ijms23073475_

Round 1

Reviewer 1 Report

Authors proposed a communication entitled “Biosynthesis of rubellins in Ramularia collo-cygni – genetic ba-2 sis and pathway proposition” for the publication in International Journal of Molecular Sciences, MDPI.

The paper has a good scientific soundness, considering its deep study of the literature and, also, in terms of the potential applications provided.

However, the addition of a paragraph at the end of the introduction section could justify in a clearer manner what is or what are the aims of this work.

My personal opinion is that the paper deserves to be published after minor revisions. 

Here is a list of some minor observations:

Figure 2. Authors could try to improve the focus of this figure: diagram is clear, but written part is not.

Line 126. “hereafter”. Should it be “hereinafter”?

Line 129. “BLASTp”. Check if this has been defined before. Maybe, this could be inserted into the abbreviation list at the end of the paper. I think that this list should be larger, due to the high number of acronyms and abbreviations reported in this work.

Line 241. “Dussart et al” Does “et al” need to be in italique? please check the guidelines of this journal.

Line 346. Was “rubellin D 4” correctly defined before? It first appeared on Line 193, but needs to be defined.

Line 386. Is this protocol verified only by Dussart et al [25], or are any other references?

Thank you

Reviewer 2 Report

The manuscript ijms-1639251 entitled "Biosynthesis of rubellins in Ramularia collo-cygni – genetic basis and pathway proposition" highlighted the biosynthetic gene cluster of rubellin in Ramularia collo-cygni. The topic fits within the scope of the journal. Overall, the manuscript is well written, easy to follow, with very nice illustrative and helpful figures. Some minor corrections that authors need to review.

Materials and Methods

L388: Is the R. collo-cygni isolate I20? Please add.

Results

L96 to L118: All the species names listed in the section 2.1. “Phylogenetic analysis of R. collo-cygni non reducing-polyketide synthases” are not in italics.

L151 to L156: This part belongs to the caption of Figure 3? If yes, please change.

Figure 5: The concentration units are different in Fig. 5A and 5B. Please change to μM.

L205 to L209: This part belongs to the caption of Figure 5. I think it was a formatting problem.
